# Systemic evaluation and localization of resistin expression in normal human tissues by a newly developed monoclonal antibody

Qing Lin[1], Shari A. Price[2], John T. Skinner[1], Bin Hu[1], Chunling Fan[1], Kazuyo Yamaji-Kegan[1], Roger A. Johns[1]*

1 Department of Anesthesiology and Critical Care Medicine, Johns Hopkins University School of Medicine, Baltimore, MD, United States of America, 2 Charles River Laboratories, Inc., Frederick, MD, United States of America

* rajohns@jhmi.edu

**Data Availability Statement:** All relevant data are within the manuscript and its Supporting Information files.

## Abstract

Resistin and resistin-like molecules are pleiotropic cytokines that are involved in inflammatory diseases. Our previous work suggested that resistin has the potential to be used as a biomarker and therapeutic target for human pulmonary arterial hypertension. However, data are limited on the distribution of resistin in healthy human organs. In this study, we used our newly developed anti-human resistin (hResistin) antibody to immunohistochemically detect the expression, localization, and intracellular/extracellular compartmentalization of hResistin in a full human tissue panel from healthy individuals. The potential cross reactivity of this monoclonal anti-hResistin IgG1 with normal human tissues also was verified. Results showed that hResistin is broadly distributed and principally localized in the cytoplasmic granules of macrophages scattered in the interstitium of most human tissues. Bone marrow hematopoietic precursor cells also exhibited hResistin signals in their cytoplasmic granules. Additionally, hResistin labeling was observed in the cytoplasm of nervous system cells. Notably, the cytokine activity of hResistin was illustrated by positively stained extracellular material in most human tissues. These data indicate that our generated antibody binds to the secreted hResistin and support its potential use for immunotherapy to reduce circulating hResistin levels in human disease. Our findings comprehensively document the basal expression patterns of hResistin protein in normal human tissues, suggest a critical role of this cytokine in normal and pathophysiologic inflammatory processes, and offer key insights for using our antibody in future pharmacokinetic studies and immunotherapeutic strategies.

## Introduction

Resistin, first identified as an adipokine in mice with insulin resistance properties [1], is a member of the resistin-like molecule (RELM) family. To date, four rodent and two human isoforms of this family have been discovered [2–5]. Members of the RELM family are pleiotropic cytokines that are involved in a variety of diseases and have distinct tissue distribution [6].

**Funding:** This work was supported by the National Institutes of Health (NIH) Centers for Advanced Diagnostics and Experimental Therapeutics in Lung Diseases (CADET) Grant NHLBI 5UH2HL123827 and NIH 5R01HL138497 (PI: R.A.J). Charles River Laboratories, Inc. performed the immunopathological study for the cross reactivity determination of anti-hResistin antibody in normal human tissues. Charles River Laboratories, Inc. also provided support for this study in the form of salary for author SAP. The specific roles of these authors are articulated in the author contributions section. Charles River Laboratories, Inc. had no role in decision to publish or preparation of the manuscript.

**Competing interests:** The authors have read the journal's policy and have the following potential competing interests: author SAP is a paid employee of Charles River Laboratories, Inc., a contract research organization that performed the immunopathological study to determine the tissue cross reactivity for the generated antihResistin antibody. The authors would like to declare the following patent applications associated with this research: RAJ had US (US 2016/0130341 A1) and international (WO 2014/204941 A1) patent applications pending for the monoclonal antibody developed against human resistin to cover pulmonary, cardiac, and other related inflammatory disorders. All other authors have declared that no competing interests exist. This does not alter our adherence to PLOS ONE policies on sharing data and materials.

Adipocytes are the main source of rodent resistin [1, 7], but mature human adipocytes were reported to lack resistin expression [8]. Under normal conditions, human resistin (hResistin) mRNA (hRETN) expression is seen predominantly in immune cells, including bone marrow cells, monocytes, and leukocytes [9]. Thus, hResistin is more related to the mouse RELMα, which is a marker of alternatively activated (M2) macrophages [10] and expressed predominantly in lung, bone marrow, and spleen [4, 5, 11, 12]. Intriguingly similarity between rodent RELMα and hResistin is not limited to their tissue and cellular sources, but also their proinflammatory and pro-proliferative functions [2, 13]. Based on data from our lab and others, hResistin and rodent RELMα have potent mitogenic, angiogenic, vasoconstrictive, and chemokine effects in lung tissue [2, 14, 15], contributing etiologically to the development of pulmonary arterial hypertension (PAH) [13, 16–18]. The proinflammatory properties of hResistin, and its potential to be used as a biomarker and therapeutic target of human PAH and other related cardiovascular inflammatory diseases have been widely studied. However, the literature lacks a systematic evaluation of the protein's expression and localization.

Therefore, we comprehensively mapped hResistin expression—and its potential production site(s)—in healthy human tissues. An expansive description of the spatial distribution of this human mediator in normal tissues could provide a framework for understanding its function in normal human cell biology and tissue homeostasis. Such information could also be used to understand how hResistin expression may be altered in human pathology, especially in cardiovascular disorders.

Given the complexity of obtaining an appropriate antibody for labeling the RELM members [6], we recently developed an anti-hResistin monoclonal antibody with functional blocking activities [19]. Using this newly developed antibody, we also established an immunohistochemical staining procedure for documenting the expression pattern of hResistin in healthy human tissues. The goal of this study was to validate our antibody as a tool for immunolabeling hResistin in disease samples. Because our anti-hResistin antibody was also developed as a therapeutic drug for the treatment of vascular inflammatory diseases, this current tissue cross-reactivity study will address the human relevance for toxicity and safety testing.

## Materials and methods

### Production of recombinant hResistin protein

We produced hResistin in a eukaryotic cell line [6]. Briefly, the pcDNA5/FRT/TOPO TA vector containing C-terminal FLAG-tagged hResistin cDNA was integrated into the genome of the Flp-In™ T-REx™ 293 cell line in a Flp recombinase-dependent manner (Invitrogen, Carlsbad, CA). Production of recombinant (r) hResistin in T-REx 293 cells was induced by 1 μg/mL tetracycline. hResistin protein then was purified from the cell culture medium by anti-FLAG M2 antibody agarose (Sigma, St. Louis, MO) column chromatography. To determine the purity of eluted hResistin proteins, SDS-PAGE were employed using 4–20% Criterio Tris-HCl protein gel (#3450033, Bio-Rad, Hercules, CA) and Coomassie (#1610436, Bio-Rad) staining. We then stimulated 3T3-L1 embryonic fibroblasts (ATCC® CL-173™, ATCC, Manassas, VA) with hResistin at different doses for 10 minutes and lysed the cells with Laemmli sample buffer. We analyzed the cell lysates by immunoblotting with anti-phospho-Akt [2, 18, 20] (#4060, Cell Signaling Technology, Danvers, MA) and anti-GAPDH (G8795, Sigma-Aldrich) to determine the activity of the purified hResistin. Western blot analysis was performed as previously described [18]. The Trans-Blot Turbo Nitrocellulose Transfer Kit (#1704271, Bia-Rad) were used and protein bands were visualized by chemiluminescence (ECL; RPN2106, GE Healthcare, Marlborough, MA).

### Anti-hResistin antibody development

Anti-hResistin antibodies were developed in cooperation with our commercial partners, Creative Biolabs (Shirley, NY) and Lonza (London, England). Antibodies were identified through a phage display approach with several human antibody libraries. Screening steps included (1) searching the phage display human scFv library, (2) ELISA and BIACORE (GE Healthcare, Pittsburgh, PA) plasmon resonance assays to determine potency and selectivity, and (3) *in silico* evaluation for immunogenicity predictions. We used an *in vitro* human smooth muscle cell proliferation bioassay to test and rank antibody function [19].

### Sample collection and analysis

Tissues collected as surgical or autopsy specimens from humans were obtained from three principal suppliers including Cooperative Human Tissue Network (Charlottesville, VA), National Disease Research Interchange (Philadelphia, PA), and Cureline (South San Francisco, CA). The Institutional Review Board (IRB) responsibility for the use of human tissues for tissue cross-reactivity studies devolved to these suppliers, which have their own IRB protocols. Additionally, the IRB acknowledged that in addition to its own governance, the conditions for the receipt of these human tissues were in accordance with the US Department of Health and Human Services regulations for the protection of human subjects (45 CFR Part 46). Before the start of this research, we obtained Ethics Committee approval for the use of human tissues in tissue cross-reactivity studies. Documentation of informed consent was required by this statement. Thus, informed consent was documented by the use of a written consent form approved by the IRB and was signed by the subject or the subject's legally authorized representative. A copy was given to the signatory of the form. Unfixed tissues as received from the suppliers were considered essentially normal. Documents indicating tissue source and any other pertinent information provided by the tissue suppliers are maintained at Charles River Laboratories (Frederick, MD).

Samples from at least three separate donors were evaluated. The tissue panel (Table 1) used as the test system included all of the tissues on the "suggested list of human tissues to be used for immunohistochemical or cytochemical investigations of cross reactivity of monoclonal antibodies" in Annex I of the European Medicines Agency (EMA) document Guideline on

**Table 1. Normal adult human tissue panel.**

| | | |
|---|---|---|
| Adrenal gland | Heart | Salivary gland |
| Bladder (urinary) | Kidney (glomerulus, tubule) | Skin |
| Blood cells [a] | Liver | Spinal cord |
| Blood vessels (endothelium) [b] | Lung | Spleen |
| Bone marrow | Lymph node | Striated muscle (skeletal) |
| Brain–cerebellum | Ovary | Testis |
| Brain–cerebrum (cerebral cortex) | Pancreas | Thymus |
| Breast | Parathyroid | Thyroid |
| Colon (large intestine) | Peripheral nerve | Tonsil |
| Eye | Pituitary gland | Ureter |
| Fallopian tube | Placenta | Uterus–cervix |
| Gastrointestinal tract [c] | Prostate | Uterus–endometrium |

[a] Evaluated from peripheral blood smears.

[b] Evaluated from all tissues where present.

[c] Includes esophagus, small intestine, and stomach (including underlying smooth muscle).

Development, Production, Characterization and Specifications for Monoclonal Antibodies and Related Products, adopted by the Committee for Medicinal Products for Human Use. It also included all of the tissues recommended in the Food and Drug Administration's (FDA), Center for Biologics Evaluation and Research document, Points to Consider in the Manufacture and Testing of Monoclonal Antibody Products for Human Use.

### Tissue sectioning and fixation

Fresh, unfixed tissue samples were previously obtained and placed into molds, filled with Tissue-Tek® OCT Compound (Sakura Finetek USA, Inc., Torrance, CA), and frozen at -85 to -70°C until sectioning. Sections were cut at 5 μm and fixed in acetone for 10 minutes at room temperature. Just prior to staining, the slides were fixed in 10% neutral-buffered formalin for 10 seconds at room temperature. Human blood smears were frozen until fixation and staining.

### Control samples

As positive control samples, we used rhResistin-FLAG UV-resin spot slides, which were produced from lab-made protein as described above. Controls were designated as hResistin-FLAG. For negative control samples, we used human hypercalcemia of malignancy peptide, amino acid residues 1–34, UV-resin spot slides. This protein was purchased from Sigma-Aldrich and designated as PTHrP 1-34. Control slides were prepared according to testing facility (Charles River Laboratories) standard operating procedures.

### Antibody concentration selection

To optimize the concentration of the test (anti-hResistin) and control (human IgG1) antibodies for staining, we evaluated multiple concentrations of anti-hResistin IgG between 0.5 and 20 μg/mL. Anti-hResistin antibody stained the positive control rhResistin-FLAG spots at all concentrations examined, although the intensity was reduced at concentrations below 5 μg/mL. The optimal concentration was considered to be the lowest concentration to produce the maximum/plateau binding to the target antigens, or 5 μg/mL. We also chose to study a concentration of hResistin IgG 4× over the optimal concentration, or 20 μg/mL (Table 2), because it was the highest concentration that did not yield nonspecific staining of control samples or test tissues.

### Immune staining procedure

We used modifications of the methods of Tuson et al. [21], Fung et al. [22], and Hierck et al. [23] for immunohistochemistry to simultaneously eliminate the biotin, peroxidase, and fluorescein of the primary antibodies and to preclude nonspecific reactivity between the secondary labeled anti-human IgG and IgG endogenous to the tissues examined. In this method, the

**Table 2. Slide set for each tissue sample.**

| Slide # | Reagent | Study scheme | Supplier |
|---|---|---|---|
| 1 | hResistin IgG, 20 μg/mL | Test article (4X optimal concentration) | Dr. Johns' lab |
| 2 | hResistin IgG, 5 μg/mL | Test article (optimal concentration) | Dr. Johns' lab |
| 3 | Human IgG1, 20 μg/mL | Control article (4X optimal concentration) | Millipore (Cat. AG502) |
| 4 | Human IgG1, 5 μg/mL | Control article (optimal concentration) | Millipore (Cat. AG502) |
| 5 | Assay control, 0 μg/mL | Omit primary antibody | — |
| 6 | Anti-β2-microglobulin, 1 μg/mL | Tissue staining control | Charles River Laboratories |

labeled secondary antibody was allowed to attach specifically to the unlabeled test or control primary antibody by overnight incubation before it was applied to the tissue cryosections. The test or control antibodies were mixed with biotinylated F(ab')2 donkey anti-human IgG, Fcγ fragment-specific (DkαHuIgG) antibody at concentrations which achieved a primary:secondary antibody ratio of 1:1.5 on the day prior to staining. Thus, the higher concentration of test or control antibody was pre-complexed with 30 μg/mL of biotinylated DkαHuIgG, whereas the lower concentration of test or control article was pre-complexed with 7.5 μg/mL of biotinylated DkαHuIgG. Pre-complexed antibodies were incubated overnight at 4˚C. Before using the antibody on the subsequent day, we added human gamma globulins to each vial to achieve a final concentration of either 6 mg/mL (higher concentration of secondary antibody) or 1.5 mg/mL (lower concentration of secondary antibody), and antibodies were incubated for 2 hours at 4˚C.

The ABC approach was used for staining. Slides were incubated with avidin solution and then biotin solution before being blocked with a solution of 1% bovine serum albumin (BSA) + 0.5% casein + 5% normal donkey serum + 100 μg/mL ssDNA. After that, the pre-complexed primary and secondary antibodies were applied to the slides for 2 hours. Next, endogenous peroxidase was quenched by Dako peroxidase block for 5 minutes. The slides were then treated with the ABC Elite reagent for 30 minutes, followed by DAB staining for 4 minutes. All slides were rinsed with tap water, counterstained, dehydrated, and mounted. Tris-buffered saline (TBS) was used as the washing buffer, and TBS + 1% BSA served as the diluent for all antibodies and ABC reagent.

### Anti-β2-microglobulin staining procedure

We performed β2-microglobulin immunohistochemistry staining of all human tissue sections to determine the adequacy of tissue samples for evaluation. The rationale was to show that the human tissue cryosections express epitopes that can be detected by immunolabeling and thus to indicate overall suitability of the tissue for study. Tissues were fixed in acetone for 10 minutes. Just prior to staining, the slides were fixed in 10% neutral-buffered formalin for 10 seconds. Fixed cryosections underwent endogenous peroxidase quenching and blocking as described above. Then, anti-β2-microglobulin antibody was applied to the slides at 1 μg/mL for 1 hour, followed by biotinylated secondary antibody (goat anti-rabbit IgG) for 30 minutes. Finally, the sections underwent ABC reagent incubation, DAB staining, counterstaining, and mounting as stated above. Phosphate-buffered saline (PBS) was used as the washing buffer and PBS + 1% BSA served as the diluent for all antibodies and the ABC Elite reagent. All steps were conducted at room temperature.

### Immunopathology evaluation

For each sample, six concurrent slides (as shown in Table 2), including hResistin antibody-stained slides and controls, were evaluated. Each slide was examined for the presence of stained cell types or tissue elements. Each stained cell type or tissue element was identified, the subcellular/extracellular location of the staining was recorded, and the intensity of staining was assigned for each slide. We also assigned a frequency for each cell type to provide the approximate percentage of cells of that type or tissue element that were stained.

The staining intensity and frequency scales used for the evaluation of hResistin expression are listed in Table 3 and Table 4, respectively. To measure the percentage of cells and areas positively stained by IHC, we set up the scales in Table 4 based on literature-reported IHC quantitative approaches [24–28] and resistin biology, including its secretory properties [6, 10, 29, 30]. The immunopathology was carried out in accordance with Good Laboratory Practice (GLP)

**Table 3. Staining intensity scale.**

| Score | Results |
|---|---|
| Neg | Negative (no stained cells) |
| ± | Equivocal (very faint stain) |
| 1+ | Weak (light stain) |
| 2+ | Moderate (light-medium stain) |
| 3+ | Strong (medium stain) |
| 4+ | Intense (dark stain) |

regulations for nonclinical laboratory studies. The complete immunopathology evaluation that was used to determine cross reactivity of the generated anti-hResistin antibody with normal human tissues is available in S1 Table online.

## Results

### Recombinant hResistin protein and anti-hResistin antibody

SDS-PAGE analysis showed that the rhResistin protein was highly purified (Fig 1A) and was able to induce Akt phosphorylation in 3T3-L1 fibroblasts *in vitro* (Fig 1B). Among several shortlisted antibody candidates, we chose the one that most potently blocked the pro-proliferative activities of hResistin in the human smooth muscle cell assay [19]. Data on the identity, strength, purity, and composition for this selected anti-hResistin antibody are shown in Fig 1C and 1D. We validated use of this antibody for histologic detection of hResistin protein in human tissues.

### Positive, negative, and tissue staining controls

The results of positive control and negative control samples are summarized in Table 5, and representative images of stained tissue are shown in Fig 2. hResistin IgG produced weak to strong staining of the positive control, rhResistin-FLAG, at concentrations of 5 and 20 μg/mL. hResistin IgG did not specifically react with the negative control, PTHrP 1–34, at either staining concentration. The control human IgG1 did not specifically react with either the positive or negative control materials. There also was no staining of the assay control slides. The assay was specific and reproducible, as indicated by the specific reactions of hResistin IgG in all staining runs with the positive control slides, the lack of specific reactivity with the negative control slides, and the lack of reactivity of the control human IgG1. The β2-microglobulin antigen is a minor Class I determinant expressed on many cell types and is strongly expressed on endothelium. Thus, we stained separate cryosections from each human test tissue in parallel for the expression of human β2-microglobulin (a relatively ubiquitous epitope) using a

**Table 4. Criteria for evaluating staining frequency.**

| Score | Percent of stained cells of a particular cell type or tissue element |
|---|---|
| Neg | Negative (no stained cells) |
| Very rare | < 1% |
| Rare | 1–5% |
| Rare to occasional | >5–25% |
| Occasional | >25–50% |
| Occasional to frequent | >50–75% |
| Frequent | >75–100% |

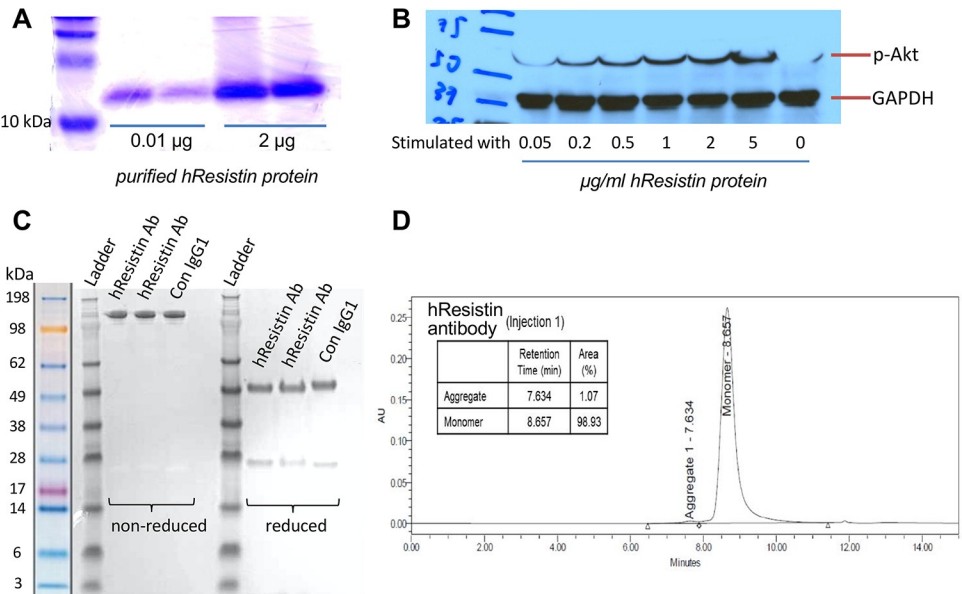

**Fig 1. Purity of the generated hResistin protein and anti-hResistin antibody.** (**A**) Coomassie-stained gel shows SDS-PAGE analysis of purified rhResistin protein. (**B**) Western blotting shows that the purified hResistin protein dose-dependently induced phosphorylation of Akt in the 3T3-L1 fibroblast cell line. (**C**) Purity of the anti-hResistin antibody and the control human IgG1 were analyzed by SDS-PAGE. (**D**) Size exclusion high-performance liquid chromatogram for the anti-hResistin antibody. Ab, antibody; Con, control; GAPDH, glyceraldehyde 3-phosphate dehydrogenase; hResistin, human resistin; p-Akt, phosphorylated Akt.

polyclonal rabbit antibody directed against human β2-microglobulin. All evaluated human tissues stained positive for β2-microglobulin (S1 Table). This finding indicates that the cryosections of human test tissues express epitopes that can be detected by immunohistochemical staining and that the tissue is suitable for inclusion in the cross-reactivity study.

## hResistin in immune system

We performed immunohistochemical staining of hResistin using the developed anti-hResistin antibody in normal human immune system tissues (Table 6 and Fig 3). In the spleen (Fig 3A), lymph nodes (Fig 3B), tonsils, and thymus (Table 6), where the immune cells accumulate, hResistin staining signals were observed in both the extracellular compartments and the cytoplasmic granules of macrophage-like immune cells. In these immune system tissues, extracellular hResistin was located in interstitium/stroma, particularly in perivascular areas. Staining often appeared as diffuse to finely punctate material and often aligned on extracellular matrix fibers. hResistin-carrying macrophages were located in the germinal centers of spleen and tonsils and were scattered in the interstitium of lymph nodes and thymus. The expression pattern of hResistin in bone marrow (Fig 3C, Table 6) was similar to that of these other immune tissues. Strong intracellular hResistin staining was also observed in the cytoplasmic granules of hematopoietic precursor cells in bone marrow.

**Table 5. Immunohistochemical analysis of the positive and negative control tissue samples.**

| | Anti-hResistin IgG | | Control human IgG1 |
|---|---|---|---|
| **Elements** | **Intensity** | **Frequency** | **Intensity/Frequency** |
| FLAG UV-resin spot (positive control) | 1–3+ | 100% | Neg |
| PTHrP 1–34 UV-resin spot (negative control) | Neg | – | Neg |

**Fig 2. Representative images of immunohistochemically stained positive and negative control samples.** (**A**) Left: positive control rhResistin protein spots stained with 5 µg/mL anti-hResistin antibody (Ab). Weak to strong test antibody staining of proteinaceous material coincident with spotted positive control rhResistin is visible. Right: no staining was apparent when hResistin protein spots were stained with 5 µg/mL control (Con) human IgG1. (**B**) Left: Negative control PTHrP 1–34 protein spots stained with 5 µg/mL anti-hResistin antibody. No specific staining is apparent. Air bubbles and minor nonspecific staining at edge of dried material are noted. Right: PTHrP 1–34 protein spots stained with 5 µg/mL human IgG1. Control antibody staining is absent. Magnification: 100×.

## hResistin in the cardiovascular-respiratory system

In the normal cardiothoracic and respiratory organ tissues (Table 7 and Fig 4), our antibody labeled secreted hResistin in the extracellular matrix of lung (Fig 4A) and heart (Fig 4B), mainly in the interstitium/stroma of perivascular areas. Staining was diffuse in these extracellular regions and aligned on extracellular matrix fibers. hResistin-positive signals were also observed intracellularly in the cytoplasmic granules of scattered macrophages in interstitium of normal lung (Fig 4A) and heart (Fig 4B). In the control IgG1-stained heart tissue, few endogenous lipofuscin pigments were observed as nonspecific background signals (Fig 4B). In the pulmonary tissue, alveolar macrophages were also a cellular source of hResistin (Fig 4A, right panels with higher magnification). hResistin-positive immune cells were rare in these thoracic organs. In human blood smears (Fig 4C), extracellular hResistin was present in serum. Few circulating blood cells were positive for hResistin.

## hResistin in the nervous system

In addition to macrophages, neuronal cell bodies/axons in several organs were shown to produce hResistin (Table 8 and Fig 5). In the nervous system, hResistin was secreted into the

**Table 6. Immunohistochemical analysis of hResistin in tissues of the immune system.**

| Tissues/Elements | Anti-hResistin IgG | | Control human IgG1 |
|---|---|---|---|
| | **Intensity** | **Frequency** | **Intensity/Frequency** |
| Spleen | | | |
| Extracellular material | 1–3+ | Occasional to frequent | Neg |
| Macrophages (cytoplasmic granules) | 1–2+ | Very rare | Neg |
| Lymph node | | | |
| Extracellular material | 1–3+ | Occasional | Neg |
| Macrophages (cytoplasmic granules) | 1–3+ | Very rare | Neg |
| Tonsil | | | |
| Extracellular material | 1–2+ | Occasional | Neg |
| Macrophages (cytoplasmic granules) | 1–2+ | Rare | Neg |
| Thymus | | | |
| Extracellular material | 1–2+ | Occasional to frequent | Neg |
| Macrophages (cytoplasmic granules) | 1–3+ | Rare | Neg |
| Bone marrow | | | |
| Extracellular material | 1–3+ | Occasional | Neg |

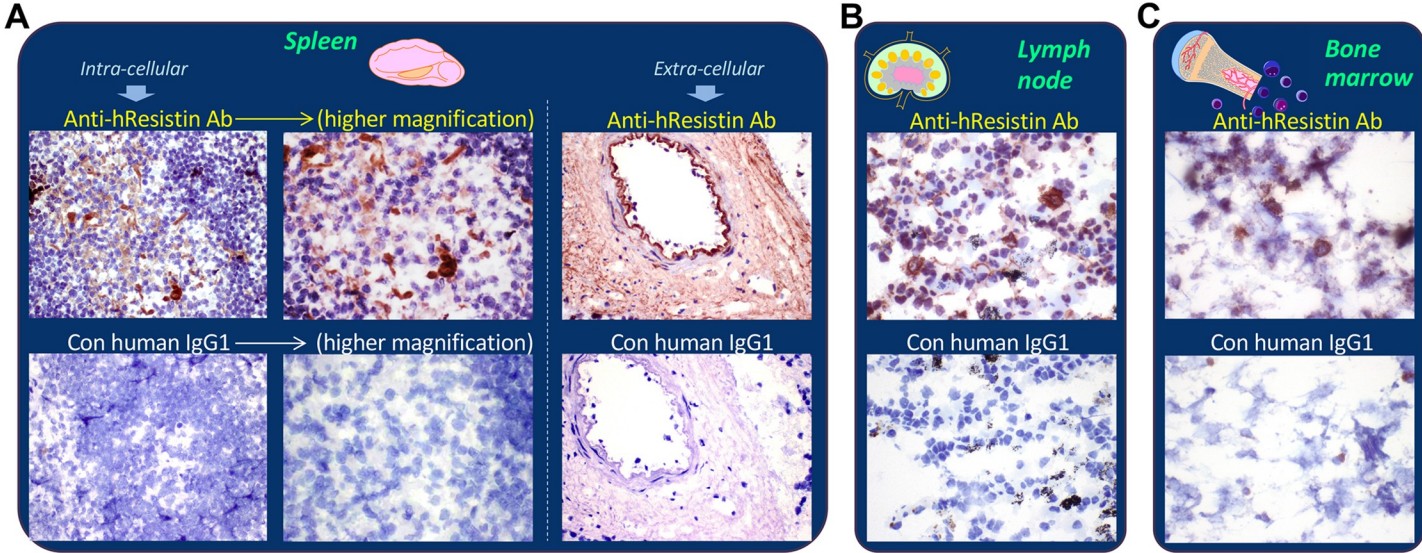

**Fig 3. Representative images of hResistin expression in the immune system.** (**A**) Human spleen tissues stained with 5 μg/mL anti-hResistin antibody. Left: anti-hResistin antibody stained cytoplasm/cytoplasmic granules in macrophages of germinal centers, whereas the same region exposed to 5 μg/mL control human IgG1 exhibited no staining. Right: anti-hResistin antibody stained the extracellular interstitium/stroma in perivascular areas, but control IgG1 did not stain any spleen tissue elements. Magnification: 400× (or higher: 600×). (**B**) Human lymph node tissues stained with 5 μg/mL anti-hResistin or control human antibodies. Magnification: 600×. (**C**) Human bone marrow tissues stained with 5 μg/mL anti-hResistin antibody or control IgG1. Magnification: 600×.

extracellular area, especially the perivascular interstitium/stroma regions. A few axons in the cerebellum expressed hResistin (Fig 5A), primarily in the Purkinje cell layer and molecular layer. In cerebrum (Fig 5B), a few neuronal cell bodies and axons were positive for hResistin, primarily in meninges. The spinal cord had a greater number of hResistin -positive axons (Table 8). Closer analysis revealed that these neuronal cell bodies and axons had similar subcellular expression patterns in which hResistin was limited to the cytoplasm. In the specimens of peripheral nerve (Table 8), more than 25–50% of cells/processes associated with peripheral nerves were positive for cytoplasmic hResistin. Comparable peripheral nerve-associated cells were observed in the spinal nerve roots of spinal cord. In the peripheral nerve tissues (Table 8), very few macrophages scattered in interstitium contained hResistin in their cytoplasmic granules.

## hResistin in the digestive system

In the tissues of the gastrointestinal tract and liver (Table 9 and Fig 6), extracellular hResistin was observed in interstitium/stroma, particularly in perivascular areas. The intracellular

**Table 7. Immunohistochemical analysis of hResistin in tissues of the cardiovascular-respiratory system.**

| Tissues/Elements | Anti-hResistin IgG | | Control human IgG1 |
|---|---|---|---|
| | **Intensity** | **Frequency** | **Intensity/Frequency** |
| Heart | | | |
| Extracellular material | 1–3+ | Occasional | Neg |
| Macrophages (cytoplasmic granules) | 1–3+ | Very rare | Neg |
| Lung | | | |
| Extracellular material | 1–3+ | Occasional to frequent | Neg |
| Macrophages (cytoplasmic granules) | 1–3+ | Very rare | Neg |
| Blood cells | | | |
| Extracellular material | 1–2+ | Frequent | Neg |

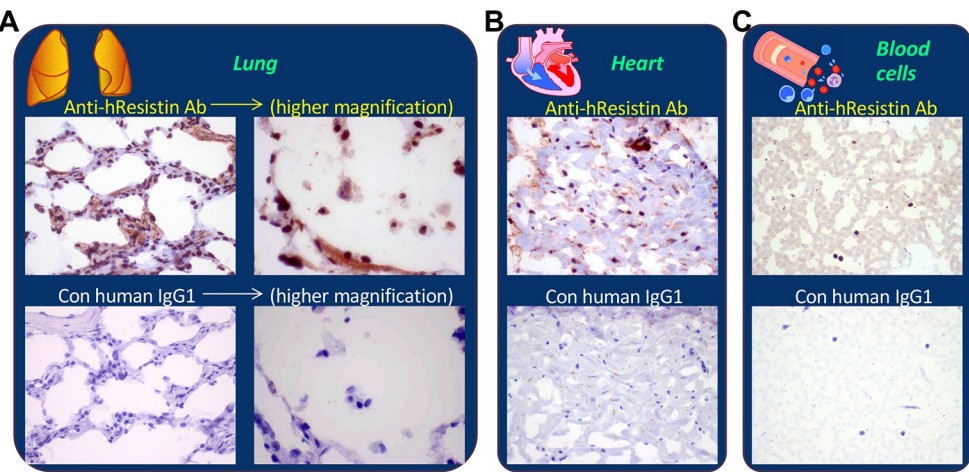

**Fig 4. Representative images of hResistin expression in the cardiovascular-respiratory system.** Human lung tissue (**A**), heart tissue (**B**), and blood cells (**C**) stained with 5 µg/mL anti-hResistin or control human IgG1 antibodies. Blood cells were evaluated from peripheral blood smears. Antibody staining was also observed in the extracellular serum. Magnification: 400× (or higher: 600× in A).

hResistin was detected in macrophages. These hResistin-expressing immune cells were scattered in interstitium of the small intestine, large intestine, esophagus, and stomach, or in lamina propria of the colon, as well as in clusters in liver (Table 9 and Fig 6). Less than 5% of observed macrophage-like myeloid cells actively expressed hResistin, and expression was limited to cytoplasmic granules of these cells in the digestive system. In stomach, 25–50% of peripheral nerve-associated cells also exhibited moderate staining by anti-hResistin antibody, and the signals were located in cytoplasm (Fig 6A).

## hResistin in the urinary system

hResistin was found in the extracellular interstitium/stroma of the kidneys, bladder, and ureters and was enriched in perivascular areas (Table 10 and Fig 7). We also observed hResistin

**Table 8. Immunohistochemical analysis of hResistin in tissues of the nervous system.**

| Tissues/Elements | Anti-hResistin IgG | | Control human IgG1 |
|---|---|---|---|
| | Intensity | Frequency | Intensity/Frequency |
| Cerebellum | | | |
| Extracellular material | 1–2+ | Occasional | Neg |
| Axons (cytoplasm) | 1–2+ | Rare | Neg |
| Cerebral cortex | | | |
| Extracellular material | 1–2+ | Occasional | Neg |
| Neuronal cell bodies/axons (cytoplasm) | 1–2+ | Rare | Neg |
| Spinal cord | | | |
| Extracellular material | 1–2+ | Occasional | Neg |
| Glial cells/processes (cytoplasm) | 1–2+ | Occasional to frequent (20 µg/ml); occasional (5 µg/ml) | Neg |
| Axons (cytoplasm) | 2–3+ | Occasional to frequent (20 µg/ml); occasional (5 µg/ml) | Neg |
| Cells/processes associated with peripheral nerves (cytoplasm) | 1–3+ | Occasional | Neg |
| Peripheral nerve | | | |
| Extracellular material | 1–2+ | Occasional | Neg |
| Cells/processes associated with peripheral nerves (cytoplasm) | 1–2+ | Occasional | Neg |
| Macrophages (cytoplasmic granules) | 1–3+ | Very rare | Neg |

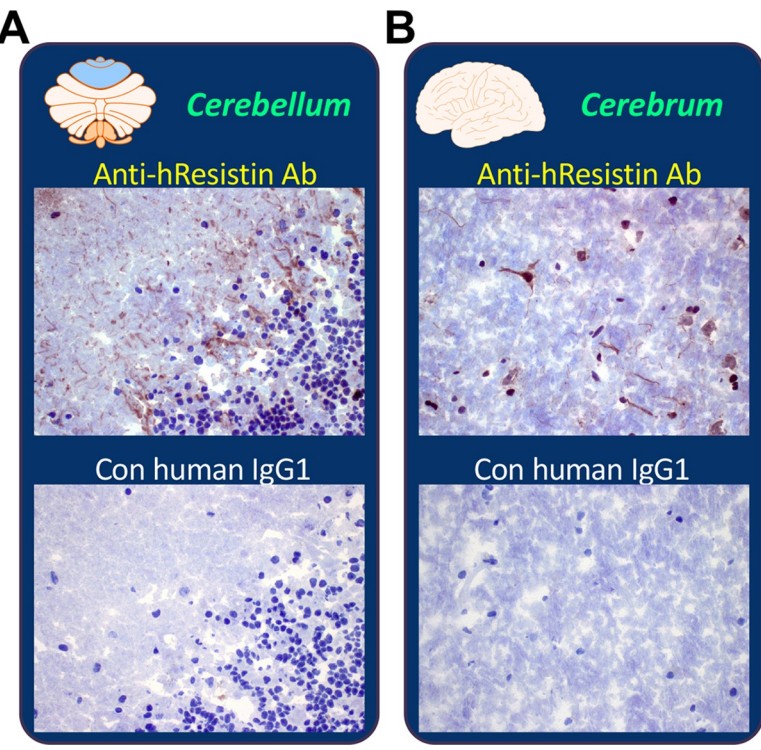

**Fig 5. Representative images of hResistin expression in the nervous system.** (**A**) Human brain cerebellum tissue stained with 5 μg/mL anti-hResistin antibody or control IgG1. Positive hResistin antibody staining was observed only rarely in the cytoplasm of axons. (**B**) Human brain cerebrum stained with 5 μg/mL anti-hResistin or control human IgG1. Cytoplasmic hResistin antibody staining of neuronal cell bodies and axons was rare. Magnification: 400×.

**Table 9. Immunohistochemical analysis of hResistin in tissues of the digestive system.**

| Tissues/Elements | Anti-hResistin IgG | | Control human IgG1 |
|---|---|---|---|
| | **Intensity** | **Frequency** | **Intensity/Frequency** |
| Small intestine | | | |
| Extracellular material | 1–2+ | Occasional to frequent | Neg |
| Macrophages (cytoplasmic granules) | 1–3+ | Rare | Neg |
| Cells/processes associated with peripheral nerves (cytoplasm) | 1–2+ | Occasional | Neg |
| Large intestine (colon) | | | |
| Extracellular material | 1–2+ | Occasional to frequent | Neg |
| Macrophages (cytoplasmic granules) | 1–3+ | Rare | Neg |
| Esophagus | | | |
| Extracellular material | 1–3+ | Occasional to frequent | Neg |
| Macrophages (cytoplasmic granules) | 1–3+ | Rare | Neg |
| Stomach | | | |
| Extracellular material | 1–2+ | Occasional to frequent | Neg |
| Macrophages (cytoplasmic granules) | 1–3+ | Rare | Neg |
| Cells/processes associated with peripheral nerves (cytoplasm) | 1–2+ | Occasional | Neg |
| Liver | | | |
| Extracellular material | 1–3+ | Occasional | Neg |
| Macrophages (cytoplasmic granules) | 1–3+ | Very rare | Neg |

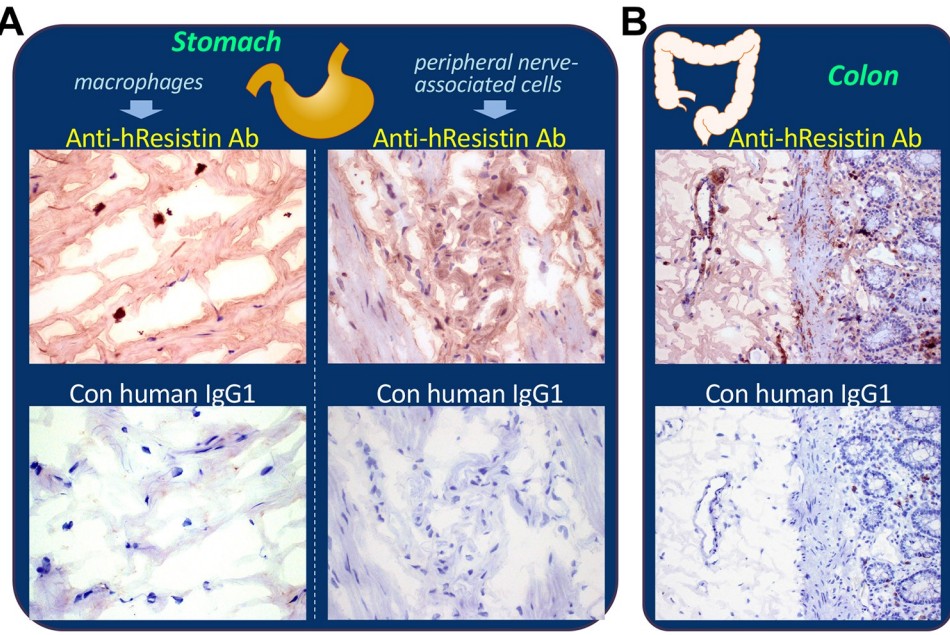

**Fig 6. Representative images of hResistin expression in the digestive system.** (**A**) Human stomach tissue stained with 5 µg/mL anti-hResistin antibody. Left: h-Resistin antibody staining of cytoplasmic granules was observed in macrophages scattered in interstitium, whereas no staining was observed in the same region with 5 µg/mL control human IgG1. Right: h-Resistin antibody staining was observed in the cytoplasm of peripheral nerve-associated cells/ processes in the myenteric plexus. Control IgG1 produced no such staining. Magnification: 400×. (**B**) Human colon tissue stained with 5 µg/mL anti-hResistin or control human IgG1. Magnification: 200×.

staining in macrophage cytoplasmic granules in the interstitium of bladders and ureters. In the kidneys (Fig 7), results were slightly inconsistent. We observed some nonspecific staining of lipofuscin granules in epithelium in two of the three individual samples. The intracellular hResistin signal was strong in a few macrophages (cytoplasmic granules) in one of the three individual kidney samples, whereas it was moderate in cytoplasm of less than 5% of peripheral nerve-associated cells in another sample. Overall, hResistin was enriched mainly in the extracellular region of kidney tissue, and intracellular staining seemed to be the exception.

**Table 10. Immunohistochemical analysis of hResistin in tissues of the urinary system.**

| Tissues/Elements | Anti-hResistin IgG | | Control human IgG1 |
|---|---|---|---|
| | **Intensity** | **Frequency** | **Intensity/Frequency** |
| Kidney (glomerulus, tubule) | | | |
| Extracellular material | 1–3+ | Occasional to frequent | Neg |
| Cells/processes associated with peripheral nerves (cytoplasm) | 1–2+ | Rare | Neg |
| Macrophages (cytoplasmic granules) | 1–3+ | Rare | Neg |
| Bladder (urinary) | | | |
| Extracellular material | 1–3+ | Occasional to frequent | Neg |
| Macrophages (cytoplasmic granules) | 1–3+ | Rare | Neg |
| Ureter | | | |
| Extracellular material | 1–3+ | Occasional to frequent | Neg |
| Macrophages (cytoplasmic granules) | 1–3+ | Occasional | Neg |

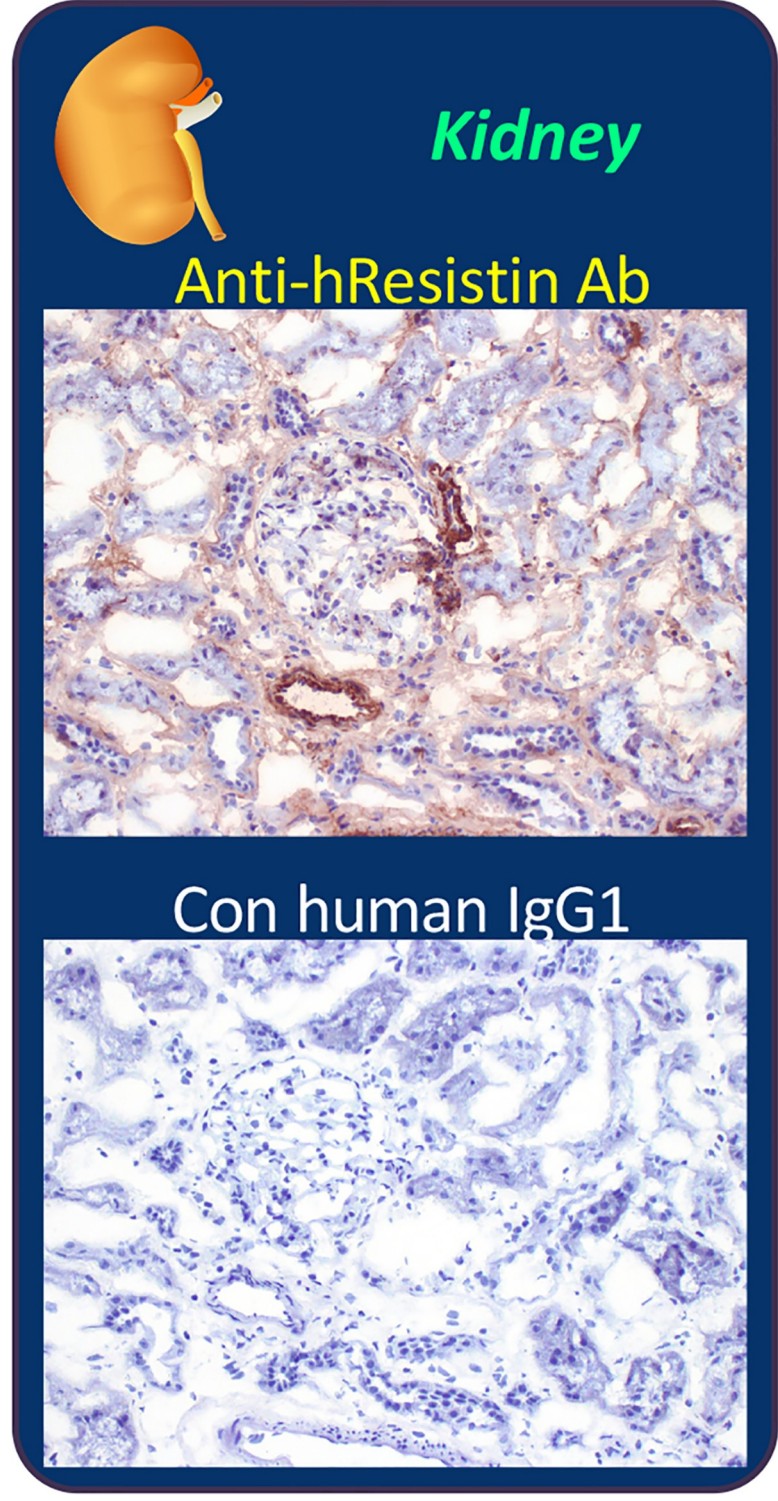

**Fig 7. Representative images of hResistin expression in the urinary system.** Human kidney tissues were stained with 5 µg/mL anti-hResistin antibody (upper) or control IgG1 (lower). In the anti-hResistin antibody-stained tissue, extracellular positive signal was observed in interstitium/stroma perivascular areas. Anti-hResistin antibody did not stain vascular endothelium in the renal tissues. Control human IgG1 did not produce any staining in kidney tissue. Magnification: 200×.

**Table 11. Immunohistochemical analysis of hResistin in tissues of the reproductive system.**

| Tissues/Elements | Anti-hResistin IgG | | Control human IgG1 |
| --- | --- | --- | --- |
| | **Intensity** | **Frequency** | **Intensity/Frequency** |
| Uterus–cervix | | | |
| Extracellular material | 1–3+ | Occasional to frequent | Neg |
| Macrophages (cytoplasmic granules) | 1–3+ | Very rare | Neg |
| Uterus–endometrium | | | |
| Extracellular material | 1–3+ | Occasional to frequent | Neg |
| Macrophages (cytoplasmic granules) | 1–3+ | Very rare | Neg |
| Ovary | | | |
| Extracellular material | 1–3+ | Occasional to frequent | Neg |
| Macrophages (cytoplasmic granules) | 1–3+ | Very rare | Neg |
| Fallopian tube | | | |
| Extracellular material | 1–3+ | Occasional to frequent | Neg |
| Macrophages (cytoplasmic granules) | 1–3+ | Rare | Neg |
| Placenta | | | |
| Extracellular material | 1–3+ | Occasional to frequent | Neg |
| Prostate | | | |
| Extracellular material | 1–3+ | Occasional to frequent | Neg |
| Cells/processes associated with peripheral nerves (cytoplasm) | 2–3+ | Rare to occasional | Neg |
| Macrophages (cytoplasmic granules) | 1–3+ | Rare | Neg |
| Testis | | | |
| Extracellular material | 1–3+ | Occasional to frequent | Neg |
| Macrophages (cytoplasmic granules) | 1–3+ | Rare | Neg |
| Mesothelium (cytoplasm, cytoplasmic granules) | 2–3+ | Frequent | Neg |
| Cells/processes associated with peripheral nerves (cytoplasm) | 1–3+ | Rare | Neg |

## hResistin in the reproductive system

Similar to our findings in other tissues, we observed extracellular hResistin signals in interstitium/stroma of the reproductive system, particularly in perivascular areas (Table 11 and Fig 8). hResistin staining was occasionally present in cytoplasmic granules of macrophages scattered in interstitium of the uterine cervix, endometrium, fallopian tubes, prostate, and testes. No hResistin-expressing macrophages were seen in placenta or ovaries, although in one ovarian

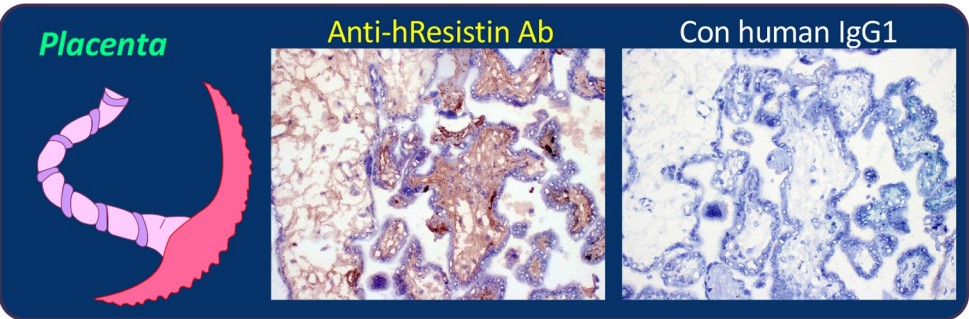

**Fig 8. Representative images of hResistin expression in the reproductive system.** Human placenta tissues stained with 5 µg/mL anti-hResistin antibody (left) or control IgG1 (right). Positive staining of secreted hResistin protein was observed in interstitium/stroma and aligned on extracellular matrix fibers. Control antibody did not produce any positive staining in human placenta. Magnification: 200×.

**Table 12. Immunohistochemical analysis of hResistin in tissues of the endocrine system.**

| Tissues/Elements | Anti-hResistin IgG | | Control human IgG1 |
|---|---|---|---|
| | Intensity | Frequency | Intensity/Frequency |
| Adrenal gland | | | |
| Extracellular material | 1–3+ | Occasional to frequent | Neg |
| Cells/processes associated with peripheral nerves (cytoplasm) | 1–3+ | Very rare | Neg |
| Macrophages (cytoplasmic granules) | 1–3+ (20 μg/mL); 1–2+ (5 μg/mL) | Rare | Neg |
| Pituitary gland | | | |
| Extracellular material | 1–3+ | Occasional | Neg |
| Epithelium, adenohypophysis (cytoplasm, cytoplasmic granules) | Neg | - | 1–3+ (occasional to frequent) |
| Macrophages (cytoplasmic granules) | 1–3+ | Very rare | Neg |
| Salivary gland | | | |
| Extracellular material | 1–3+ | Occasional to frequent | Neg |
| Macrophages (cytoplasmic granules) | 2–3+ | Very rare | Neg |
| Thyroid | | | |
| Extracellular material | 1–2+ | Occasional to frequent | Neg |
| Parathyroid | | | |
| Extracellular material | 1–2+ | Occasional | Neg |
| Macrophages (cytoplasmic granules) | 1–3+ | Rare | Neg |
| Pancreas | | | |
| Extracellular material | 1–3+ | Occasional | Neg |
| Cells/processes associated with peripheral nerves (cytoplasm) | 1–3+ | Occasional | Neg |
| Macrophages (cytoplasmic granules) | 1–3+ | Very rare | Neg |

sample fewer than 1% of immune cells were positive for hResistin. Interestingly, in one of the three individual testis samples, hResistin IgG also produced moderate to strong staining of the cytoplasm and cytoplasmic granules in many mesothelial cells. Less than 5% of peripheral nerve-associated cells stained positively for hResistin in cytoplasm of that mesothelium-stained testis sample.

## hResistin in the endocrine system

We detected extracellular hResistin staining in interstitium/stroma, particularly in perivascular areas, of endocrine organs including adrenal gland, pituitary gland, salivary gland, thyroid, para-thyroid, and pancreas (Table 12 and Fig 9). hResistin was also secreted into the colloid of the thyroid (Table 12). Macrophages exhibiting strong hResistin signals in normal pituitary and salivary glands were very rare, less than 1%, scattered in interstitium (Table 12). One of the three individual samples exhibited less than 5% hResistin-positive interstitium macrophages and an accumulation of positive signals in cytoplasmic granules in adrenal gland, para-thyroid, and pancreas (Table 12 and Fig 9). In pancreas (Fig 9), very few (less than 1%) macrophages exhibited strong cytoplasmic hResistin staining, but 25–50% of peripheral nerve-associated cells in the pancreas showed staining for hResistin (Fig 9). Very little nonspecific staining was observed in these endocrine tissues, although some nonspecific staining was observed in pigment granules of the adrenal glands and some nonspecific background staining was present in epithelium cytoplasm of adenohypophysis in the pituitaries. This nonspecific staining did not hamper interpretation of the results as similar staining was not observed with the anti-hResistin antibody. The overall histologic evaluation showed a predominately secreted expression pattern for hResistin in normal human endocrine system.

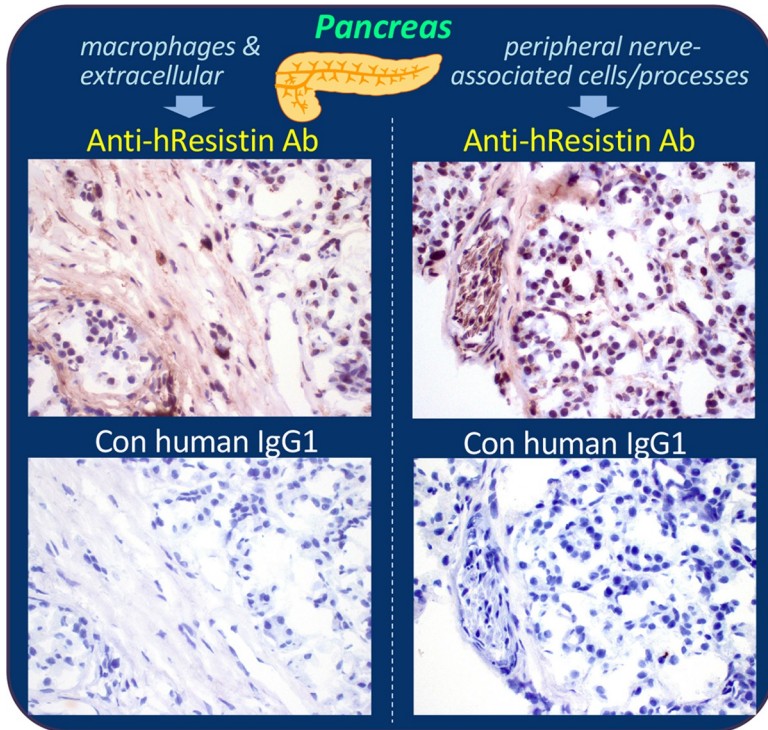

**Fig 9. Representative images of hResistin expression in the endocrine system.** Human pancreas tissues stained with 5 µg/mL anti-hResistin antibody (upper panels) or control IgG1 (lower panels). Left: positive staining for hResistin was located in cytoplasmic granules of very rare macrophages scattered in interstitium, and in extracellular interstitium/ stroma. Right: hResistin staining was also observed in cytoplasm of occasional cells/processes associated with peripheral nerves in some areas of pancreatic tissue. These regions did not exhibit staining by control human IgG1. Magnification: 400×.

## hResistin in integumentary/muscular/sensory systems

In the tissues of skin (Fig 10A) and striated muscle (Fig 10B), extracellular interstitium/stroma, particularly the perivascular areas, were positively stained by the anti-hResistin antibody. Staining was aligned on extracellular matrix fibers. Rarely occurring macrophages showed hResistin signals in cytoplasmic granules. In breast tissues (Fig 10C), hResistin had an expression pattern similar to that in skin and skeletal muscle. It was observed in extracellular matrix and in the cytoplasmic granules of scattered macrophages in interstitium of mammary glands. In eye tissue (Fig 10D), endogenous melanin pigments could be seen in retina, choroid, sclera, iris, ciliary body/processes, extraocular muscle, cornea, conjunctiva, and lens. Moderate extracellular hResistin signals were detected in interstitium/stroma. The hResistin staining smeared over multiple tissue elements but was particularly prominent in sclera, retina, and ciliary body. hResistin IgG also labeled less than 1% of macrophages in interstitium of conjunctiva, mainly in the subcellular cytoplasmic granules (Table 13 and Fig 10D).

## Overall distribution mapping of hResistin protein expression in human tissues

We summarize the overall body map of hResistin distribution in normal human tissues by a schematic illustration in Fig 11. Comprehensive evaluation with hResistin IgG revealed that macrophages scattered in interstitium are the main cell type to produce hResistin. Hematopoietic precursor cells, an immune cell population in bone marrow, also secrete hResistin.

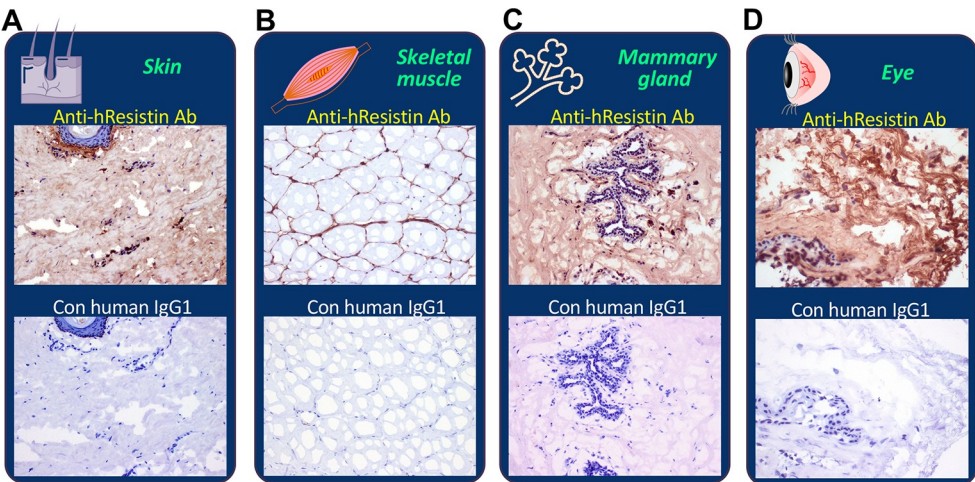

**Fig 10. Representative images of hResistin expression in the integumentary/muscular/sensory systems.** Tissues of human skin (A), striated skeletal muscle (B), mammary gland/breast (C), and eyes (D) were stained with 5 µg/mL anti-hResistin antibody or control human IgG1. Magnification: 200× (A, B, and D) or 400× (C).

Neuronal cells are a third cellular source of hResistin. Neuronal cells, axons, and glial cells in brain and/or spinal cord, as well as peripheral nerve-associated cells in the organs of digestive, urinary/reproductive, nervous, and endocrine systems were positive for hResistin. hResistin was localized to the cytoplasm of all the hResistin-expressing cells. We observed no cell membrane staining. Extracellular regions in most normal human tissues were also positive for hResistin, reflecting the binding of the antibody to secreted and circulating hResistin.

## Discussion

In this study, we characterized hResistin expression across normal human tissues using a newly developed monoclonal antibody. Given the complexity and difficulty of choosing the appropriate antibodies for labelling RELM proteins [6], generating a specific anti-hResistin antibody as a reliable and specific detecting tool was crucial for this study. Our antibody explicitly detects hResistin protein, as evident from its immunoreactivity in a panel of normal

**Table 13. Immunohistochemical analysis of hResistin in tissues of the integumentary/muscular/sensory systems.**

| Tissues/Elements | Anti-hResistin IgG | | Control human IgG1 |
|---|---|---|---|
| | **Intensity** | **Frequency** | **Intensity/Frequency** |
| Striated muscle (skeletal) | | | |
| Extracellular material | 1–3+ | Occasional | Neg |
| Macrophages (cytoplasmic granules) | 1–3+ | Very rare | Neg |
| Skin | | | |
| Extracellular material | 1–3+ | Occasional | Neg |
| Macrophages (cytoplasmic granules) | 2–3+ | Very rare | Neg |
| Breast | | | |
| Extracellular material | 1–2+ | Occasional | Neg |
| Macrophages (cytoplasmic granules) | 1–3+ | Rare | Neg |
| Eye | | | |
| Extracellular material | 1–2+ | Occasional to frequent | Neg |
| Macrophages (cytoplasmic granules) | 1–3+ | very rare | Neg |

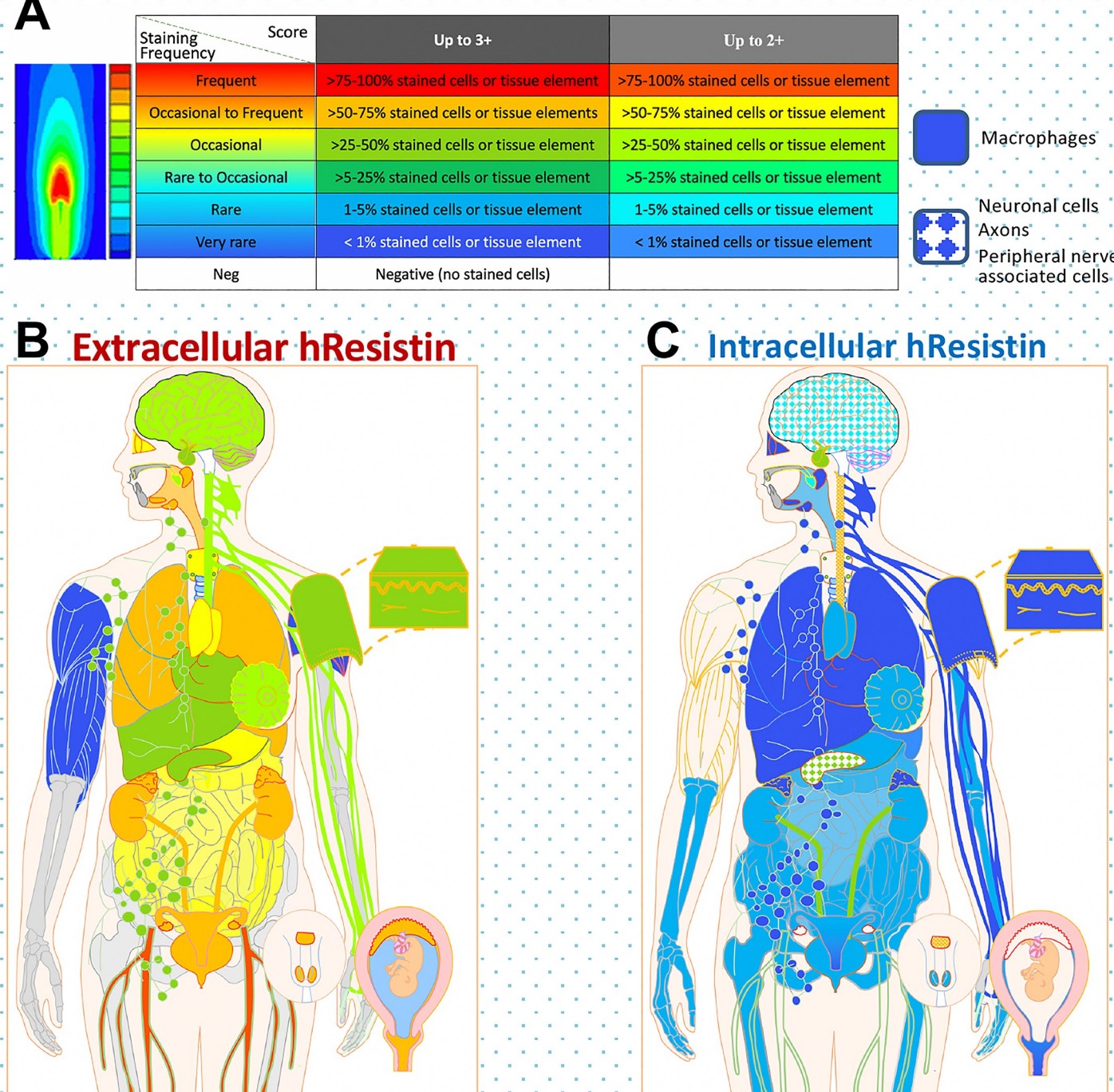

**Fig 11. Global analysis of hResistin expression in normal human organ tissues.** (**A**) Color coding (from cold [blue] to warm [red]) designates the positive intensity and frequency of anti-hResistin antibody staining. For the analysis of intracellular hResistin distribution, solid color-filled areas indicate the presence of hResistin-producing macrophages, whereas the pattern denotes hResistin-expressing peripheral nerve-associated cells, neuronal cells, or axons in the tissues. (**B** and **C**) Global mapping of the expression and distribution of extracellular (B) and intracellular (C) hResistin in normal human tissues.

human tissues. Moreover, our results from several positive, negative, and assay controls in the panel of human tissues clearly portrayed the specificity of our antibody with no nonspecific cross reactivity.

We used immunohistochemistry to evaluate hResistin expression because it enabled us to determine the proportion of cells exhibiting hResistin, the protein's intracellular and extracellular localization, and the level and pattern of expression in human tissues [31, 32]. hResistin has been linked to human inflammatory diseases in multiple organ systems, including endotoxemia, sepsis, rheumatoid arthritis, and inflammatory bowel disease [33, 34]. Studies of the contribution of this cytokine to the pathogenesis of cardiovascular diseases, especially human PAH, are beginning to expand rapidly. hResistin is thus emerging as a potential biomarker and therapeutic target for cardiothoracic diseases. As reported, hResistin is frequently upregulated under pathological states [5, 35], suggesting that successful targeting of this protein to treat PAH is likely to prove highly specific and to have few side effects. Hence, we developed therapeutic antibodies against hResistin and evaluated their effectiveness and initial feasibility for human use by using preclinical *in vitro* and *in vivo* models [19]. Using this antibody, we further provide documentation of hResistin's expression pattern in healthy human organs. The data provided will contribute to our understanding of the normal function of this protein and to interpretation of its altered expression in human vascular inflammation-related diseases.

In the human tissue panel tested, we observed no cell membrane staining. The main cellular sources of hResistin were macrophages, although these immune cells were uncommon in healthy tissues. hResistin IgG produced strong staining in cytoplasmic granules of the macrophages, which were primarily scattered in the interstitium of most tissues. hResistin-expressing macrophages were highly present in immune system tissue, including spleen and lymph nodes, in line with a previous report in the literature [6]. Hematopoietic precursor cells in bone marrow also were positive for hResistin, further validating the interaction between hResistin and stem/progenitor cells that we have previously reported [20, 36]. Interestingly, neuronal cell bodies/axons in brain and spinal cord, as well as a few peripheral nerve-associated cells in the gastrointestinal tract, endocrine system, and renal/reproductive system also produced hResistin. Under neuropathologic conditions, hResistin has been shown to inhibit mitochondrial biogenesis of human neuronal cells *in vitro* by suppressing PGC-1α and AMPK pathways [37]. Secreted hResistin also has been found in human cerebrospinal fluid (CSF). In healthy individuals, CSF hResistin is generally low, but various neurologic diseases can elevate its expression. Hence, CSF-derived hResistin might represent an inflammatory marker of these diseases [38, 39]. Of note, our anti-hResistin antibody also stained extracellular hResistin in most human tissues and the circulating hResistin in serum of human blood smears. The fact that hResistin is secreted makes it easily accessible to antibodies in the circulation. Hence it could be used as a target for therapy similar to other cytokines and chemokines. As a cytokine/chemokine-like protein, hResistin has multiple potential receptors, but its main receptor(s) remains unidentified. Therefore, direct targeting of a receptor is not an option at this time, supporting the rationale for using monoclonal antibody-based therapy.

Our tissue cross-reactivity study is also required for the further development of a therapeutic anti-hResistin antibody. According to FDA and EMA suggestions, monoclonal antibodies should be assessed for binding to target and possible non-target human tissues. The *in vitro* evaluation of cross reactivity in tissue specimens may identify potential tissue sites or organ systems that should be evaluated more thoroughly in subsequent preclinical studies to provide support for the possible future use of hResistin IgG in humans. The β2-microglobulin staining data qualified the human tissue sections for cross-reactivity evaluation. The rationale was to demonstrate that these cryosections express epitopes that can be detected by immunohistochemical staining and to indicate overall tissue suitability for testing. These cross-reactivity

data will help to establish patterns of on- and off-target tissue binding and form the basis for further toxicology/safety and pharmacokinetic studies for this antibody in humans. The results indicated that binding of our monoclonal antibody to cytoplasmic sites generally has little to no toxicologic significance, as antibody drugs are unable to access the cytoplasmic compartment *in vivo* [40]. The toxicologic significance of the extracellular binding is still unclear and requires further study.

Our study had several limitations. First, the main cellular source of hResistin was macrophage-like immune cells. Co-localization studies with specific macrophage/monocyte marker (s) are needed to confirm the hResistin-expressing macrophages in normal human tissues. Second, although it is unlikely, we cannot exclude the possibility that some of the hResistin-positive cells may be hResistin-binding rather than hResistin-producing immune cells. Fluorescence *in situ* hybridization (FISH) negative staining for hResistin mRNA (hRETN) in tissues would help to distinguish the hResistin-targeted cells from the hResistin-producing cells. In addition, the "normal condition" in our study is not absolute, as a small amount of hypoxia may be present in the tissues collected from autopsy specimens that might also cause slight recruitment of hResistin-positive macrophages and secretion of the extracellular hResistin. Future studies with tissues from other sources may help to address this limitation.

In sum, our study, which used a newly produced anti-hResistin antibody, defined the basal protein expression of hResistin in normal human tissues and provided a foundation for the recognition and interpretation of changes in these patterns that may be associated with disease states. Widespread tissue distribution of hResistin indicates a natural biological function for this inflammatory cytokine and suggests an intriguing complexity of hResistin-associated pathologies given the involvement of multiple related body systems. Under pathologic conditions, hResistin is likely to be regulated spatially in a cell type- and tissue-specific manner. Each tissue or cell type may also display a unique physiologic response to hResistin. Thus, in future projects, the involvement of hResistin in specific disorders needs to be clarified in each specific tissue. The anti-hResistin antibody that we developed as a histologic immunolabeling tool for the study of human tissues may be applicable to such studies. Additional research is needed to address whether this tool can be applied to other assays such as western blotting or ELISA for quantitative hResistin expression analysis in humans.

## Supporting information

**S1 Table. Immunopathology evaluation: Cross reactivity of hResistin IgG with normal human tissues.**
(PDF)

**S1 Raw Images.**
(PDF)

## Acknowledgments

We thank Claire F. Levine, MS, ELS, for editing this article in manuscript form. We also would like to thank all of our collaborators and contractors for their indispensable efforts and contributions in the development of the monoclonal anti-hResistin antibody.

## Author Contributions

**Conceptualization:** Roger A. Johns.

**Data curation:** Shari A. Price, John T. Skinner.

**Formal analysis:** Shari A. Price.

**Funding acquisition:** Roger A. Johns.

**Investigation:** Qing Lin, Shari A. Price, John T. Skinner, Chunling Fan, Kazuyo Yamaji-Kegan.

**Methodology:** Shari A. Price, Roger A. Johns.

**Project administration:** John T. Skinner.

**Resources:** Shari A. Price, John T. Skinner, Chunling Fan.

**Supervision:** Roger A. Johns.

**Validation:** Shari A. Price, Bin Hu, Chunling Fan.

**Visualization:** Qing Lin, Shari A. Price.

**Writing – original draft:** Qing Lin.

**Writing – review & editing:** Qing Lin, Shari A. Price, Roger A. Johns.

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
