## [Decision Letter · Decision Letter 0]

24 Mar 2020

PONE-D-20-02185

Systemic evaluation and localization of resistin expression in normal human tissues by a newly developed monoclonal antibody

PLOS ONE

Dear Dr. Johns,

Thank you for submitting your manuscript to PLOS ONE. After careful consideration, we feel that it has merit but does not fully meet PLOS ONE’s publication criteria as it currently stands. Therefore, we invite you to submit a revised version of the manuscript that addresses the points raised during the review process.

We would appreciate receiving your revised manuscript by May 08 2020 11:59PM. To enhance the reproducibility of your results, we recommend that if applicable you deposit your laboratory protocols in protocols.io, where a protocol can be assigned its own identifier (DOI) such that it can be cited independently in the future. For instructions see: http://journals.plos.org/plosone/s/submission-guidelines#loc-laboratory-protocols

We look forward to receiving your revised manuscript.

Kind regards,

Arun Rishi, Ph.D.

Academic Editor

PLOS ONE

Journal Requirements:

"Before the start of this research, the “Ethics Committee Approval for the Use of Human

Tissues in Tissue Cross-Reactivity Studies” was obtained. Documentation of informed

consent was required by this statement indicating that “informed consent shall be

documented by the use of a written consent form approved by the IRB and signed by

the subject or the subject's legally authorized representative, and a copy shall be given

to the person signing the form”.

Once you have amended this statement in the Methods section of the manuscript, please add the same text to the “Ethics Statement” field of the submission form (via “Edit Submission”).

6. Thank you for stating the following in the Financial Disclosure section:

"This work was supported by National Institutes of Health (NIH) Centers for Advanced

Diagnostics and Experimental Therapeutics in Lung Diseases (CADET) Grant NHLBI

5UH2HL123827 (PI: R.A.J)."

We note that one or more of the authors are employed by a commercial company: "Charles River Laboratories, Inc."

Reviewers' comments:

Reviewer's Responses to Questions

**Comments to the Author**

1. Is the manuscript technically sound, and do the data support the conclusions?

Reviewer #1: Yes

2. Has the statistical analysis been performed appropriately and rigorously? 

Reviewer #1: Yes

3. Have the authors made all data underlying the findings in their manuscript fully available?

Reviewer #1: Yes

4. Is the manuscript presented in an intelligible fashion and written in standard English?

Reviewer #1: Yes

5. Review Comments to the Author

Reviewer #1: Thank you for inviting me to review this manuscript. The authors are to be commended on performing a rigorous study of the distribution of human resistin in human tissues and for presenting the results in an intelligible and well-organized fashion. The manuscript is sound. I have a couple of minor comments, relating to the methodology:

-the authors should explain why they selected the antibodies used for testing (p4, l.65-67), or provide relevant references.

-the authors should explain the non-linear grading system used in table 4 (i.e. frequencies of <25% can fall within one of 3 frequency categories - very rare, rare or rare to occasional - whereas each subsequent frequency is a denomination of 25%. How and why did they select the cutoffs? Is there a supporting reference?

6. PLOS authors have the option to publish the peer review history of their article (what does this mean?). If published, this will include your full peer review and any attached files.

Reviewer #1: Yes: Anthony Bonavia

---

## [Author Response · Author response to Decision Letter 0]

1 Jun 2020

Point-by-point responses to:

Academic Editor’s Comments for Journal Requirements: 

Response: We thank the editor for this reminder. With kind help from Ms. Claire Levine, MS, ELS, who is the scientific editor in our department (Anesthesiology and Critical Care Medicine, Johns Hopkins University), the manuscript has been formatted to meet PLOS ONE's style requirements including the file naming. 

"Before the start of this research, the “Ethics Committee Approval for the Use of Human Tissues in Tissue Cross-Reactivity Studies” was obtained. Documentation of informed consent was required by this statement indicating that “informed consent shall be documented by the use of a written consent form approved by the IRB and signed by the subject or the subject's legally authorized representative, and a copy shall be given

to the person signing the form”.

Once you have amended this statement in the Methods section of the manuscript, please add the same text to the “Ethics Statement” field of the submission form (via “Edit Submission”).

Response: We greatly appreciate the editor’s advice. We have amended the statement accordingly in the Methods section of the revised manuscript (pages 6: lines 94 to 99). We also made the same changes to “Ethics Statement” field of the submission form. 

3-a) If there are ethical or legal restrictions on sharing a de-identified data set, please explain them in detail (e.g., data contain potentially identifying or sensitive patient information) and who has imposed them (e.g., an ethics committee). Please also provide contact information for a data access committee, ethics committee, or other institutional body to which data requests may be sent.

3-b) If there are no restrictions, please upload the minimal anonymized data set necessary to replicate your study findings as either Supporting Information files or to a stable, public repository and provide us with the relevant URLs, DOIs, or accession numbers. Please see http://www.bmj.com/content/340/bmj.c181.long for guidelines on how to de-identify and prepare clinical data for publication. For a list of acceptable repositories, please see http://journals.plos.org/plosone/s/data-availability#loc-recommended-repositories.

Response: We thank the editor for these instructions. We are sorry for any misunderstanding that our previous description might have caused, as actually all data underlying the findings had been presented in the originally submitted manuscript, which we believe met requirements of the PLOS-defined “minimal data set.” However, to further strengthen our findings, in the revised manuscript we provide the entire immunopathology evaluation used to determine cross reactivity of the anti-hResistin antibody with normal human tissues as supporting information (S1 Table). Now “all relevant data are within the revised paper and its supporting information files,” as we stated in the submission form and in revised manuscript (page 36). We also adjusted the related text description in the Methods section (page 11: lines 199 to 201) and addressed this concern in the revised cover letter. 

Response: The S1 Table provided as a Supporting Information file in the revised manuscript contains data to support these findings. We accordingly added a citation (S1 Table) to replace the “data not shown” in the revised manuscript (page 14: line 240). 

Response: The corresponding author, Dr. Roger A. Johns, has a pre-existing ORCID: “0000-0001-9232-2434”. It has been updated in the submission system. 

6. Thank you for stating the following in the Financial Disclosure section:

"This work was supported by National Institutes of Health (NIH) Centers for Advanced

Diagnostics and Experimental Therapeutics in Lung Diseases (CADET) Grant NHLBI

5UH2HL123827 (PI: R.A.J)."

We note that one or more of the authors are employed by a commercial company: "Charles River Laboratories, Inc."

6-a) Please provide an amended Funding Statement declaring this commercial affiliation, as well as a statement regarding the Role of Funders in your study. If the funding organization did not play a role in the study design, data collection and analysis, decision to publish, or preparation of the manuscript and only provided financial support in the form of authors' salaries and/or research materials, please review your statements relating to the author contributions, and ensure you have specifically and accurately indicated the role(s) that these authors had in your study. You can update author roles in the Author Contributions section of the online submission form.

6-b) Please also provide an updated Competing Interests Statement declaring this commercial affiliation along with any other relevant declarations relating to employment, consultancy, patents, products in development, or marketed products, etc. 

Response: We thank the editor for pointing out this matter. In the revised manuscript, we have amended the Funding Statement to clarify the role of Charles River Laboratories, Inc. in this study (page 35: lines 558 to 562). Charles River Laboratories, Inc. is not a funder. It is a contract research organization that performed the immunopathological study for determining the cross reactivity of anti-hResistin antibody in normal human tissues. The co-author Dr. Shari A. Price is an employee of Charles River Laboratories, Inc. The specific roles of these authors are articulated in the “author contributions” section (page 37) in the revised manuscript.

As advised, the Competing Interests Statement also has been updated in the revised manuscript (page 36). In addition to the clarification of the role of Charles River Laboratories, Inc., we also declared that Dr. Roger A. Johns has US (US 2016/0130341 A1) and international (WO 2014/204941 A1) patent applications pending for the monoclonal antibody developed against human resistin to cover pulmonary, cardiac, and other related inflammatory disorders. We confirmed that “This does not alter our adherence to PLOS ONE policies on sharing data and materials” at the end of the Competing Interests Statement.

Both the updated Funding Statement and Competing Interests Statement have been included in the revised cover letter. We understand that all of these amendments are required by PLOS ONE policy. 

Point-by-point responses to:

Reviewer’s Comments: 

Reviewer #1: Thank you for inviting me to review this manuscript. The authors are to be commended on performing a rigorous study of the distribution of human resistin in human tissues and for presenting the results in an intelligible and well-organized fashion. The manuscript is sound. I have a couple of minor comments, relating to the methodology:

Response: We greatly appreciate the reviewer’s general comments, and have addressed specific comments in a point-by-point manner as follows.

-the authors should explain why they selected the antibodies used for testing (p4, l.65-67), or provide relevant references. 

Response: In our previous studies, we showed that the resistin-like molecule (RELM) α exhibited pro-proliferative effects through Akt phosphorylation on rodent pulmonary smooth muscle cells (Circ Res. 2003, PMID: 12714564) and human mesenchymal stem cells (Stem Cells Dev. 2013, PMID: 22891677). Similarly we had found that both RELMα and its human homolog resistin (hResistin) also could induce Akt phosphorylation in mouse embryonic fibroblast (adipocyte precursor) 3T3-L1 cells. Thus Akt phosphorylation has been employed as an indicator for validating the functional activities of RELMα and hResistin that were produced and purified by our laboratory (J Immunol. 2019, PMID: 31611261). Relevant references are provided in the revised manuscript to support the methodology (page 5: line 70; references: 2, 18 and 20). 

-the authors should explain the non-linear grading system used in table 4 (i.e. frequencies of <25% can fall within one of 3 frequency categories - very rare, rare or rare to occasional - whereas each subsequent frequency is a denomination of 25%. How and why did they select the cutoffs? Is there a supporting reference? 

Response: We thank the reviewer for pointing out this matter. The staining frequency scales in Table 4 were based on the literature-reported IHC quantitative approaches and resistin biology. Human resistin is mainly produced by macrophages (BBRC. 2003, PMID: 12504108), and as a secretory cytokine, it has both intracellular and extracellular localization. Thus, we specifically referred to the established IHC scoring systems used to analyze targeted proteins with a similar expression pattern. The scores dichotomized at the 25% cutoff point as rare vs. variable/occasional are well-accepted for measuring the percent of cells and area positively stained by IHC (Diagn Pathol. 2014, PMID: 25432701; J Biomed Mater Res A. 2014, PMID: 23765602; PLoS One. 2014, PMID: 24802416). Those tissue elements with >25% positive resistin staining were proposed to comprise the majority of extracellularly secreted protein, whereas those with <25% positive levels were considered to have predominately intracellular expression (J Biomed Mater Res A. 2014, PMID: 23765602). In the above-listed references, sample denominations of 25% (i.e., 50%, 75%, and 100) were used for staining with >25% frequency. For those tissues with a rare level of resistin signal (<25%), we chose 1% and 5% as cut-points based on the established quantification specific for intracellular staining evaluation (Diagn Pathol. 2014, PMID: 25432701; J Clin Pathol. 1995, PMID: 7490328; Mod Pathol. 1998, PMID: 9504686). It has been difficult to find published data on the ratio of resistin-positive cells/total cells in normal human tissues. As a human homolog of rodent RELMα and an M2 macrophage marker (as discussed in page 3, line 33 with reference-10 in the revised manuscript), human resistin is likely also expressed by M2-like macrophage subsets. According to the literature, in virtually all adult mammal tissues, resident macrophages could represent up to 10% of the total cell number in quiescent conditions (Front Immunol. 2014, PMID: 25368618). The ratio of human M1 and M2 macrophages remains in a 1:1 balance under normal conditions (Exp Ther Med. 2018, PMID: 30546406). Therefore, the resistin-positive cells are likely less than 5% in normal human tissues. The resistin expression in human lung could serve as a clue because it has been implicated in pulmonary diseases (J Immunol. 2019, PMID: 31611261). In noncancerous transplanted human lungs (Sci Transl Med. 2019, PMID: 30760579), the M2-like macrophage frequency among all nucleated cells was approximately 1% to 5%, which was concordant with the detected resistin-positive macrophage-like cells in human transplanted lungs without pulmonary hypertension in our previous study (J Immunol. 2019, PMID: 31611261). Moreover, we originally had planned that these proposed scales might be modified at our expert pathologists’ discretion to better reflect the staining frequency seen during evaluation. Subsequently, the results (presented in this manuscript) turned out to strengthen the evidence for using this nonlinear grading system as an appropriate tool for resistin expression analysis. Collectively, we believe that this information could theoretically and practically support the proposed cutoffs set up as the scales in Table 4. In the revised manuscript, we added a brief discussion to clarify the rationale for this IHC quantitative approach (page 11: lines 194 to 198). Related supporting references (# 24-30) are also provided.

---

## [Decision Letter · Decision Letter 1]

18 Jun 2020

Systemic evaluation and localization of resistin expression in normal human tissues by a newly developed monoclonal antibody

PONE-D-20-02185R1

Dear Dr. Johns,

We’re pleased to inform you that your manuscript has been judged scientifically suitable for publication and will be formally accepted for publication once it meets all outstanding technical requirements.

Kind regards,

Arun Rishi, Ph.D.

Academic Editor

PLOS ONE

Additional Editor Comments (optional):

Reviewers' comments:

Reviewer's Responses to Questions

**Comments to the Author**

1. If the authors have adequately addressed your comments raised in a previous round of review and you feel that this manuscript is now acceptable for publication, you may indicate that here to bypass the “Comments to the Author” section, enter your conflict of interest statement in the “Confidential to Editor” section, and submit your "Accept" recommendation.

Reviewer #1: All comments have been addressed

2. Is the manuscript technically sound, and do the data support the conclusions?

Reviewer #1: Yes

3. Has the statistical analysis been performed appropriately and rigorously? 

Reviewer #1: Yes

4. Have the authors made all data underlying the findings in their manuscript fully available?

Reviewer #1: Yes

5. Is the manuscript presented in an intelligible fashion and written in standard English?

Reviewer #1: Yes

6. Review Comments to the Author

Reviewer #1: (No Response)

7. PLOS authors have the option to publish the peer review history of their article (what does this mean?). If published, this will include your full peer review and any attached files.

Reviewer #1: No

---

## [Editor Report · Acceptance letter]

22 Jun 2020

PONE-D-20-02185R1 

Systemic evaluation and localization of resistin expression in normal human tissues by a newly developed monoclonal antibody 

Dear Dr. Johns:

I'm pleased to inform you that your manuscript has been deemed suitable for publication in PLOS ONE. Congratulations! Your manuscript is now with our production department. 

Kind regards, 

on behalf of

Prof Arun Rishi 

Academic Editor

PLOS ONE